# Gender and Professional Role Differences in Chilean Educational Personnel’s Perceptions of School Climate and Well-Being

**DOI:** 10.3390/bs15111447

**Published:** 2025-10-24

**Authors:** Flavio Muñoz-Troncoso, Enrique Riquelme-Mella, Ignacio Montero, Gerardo Muñoz-Troncoso

**Affiliations:** 1Faculty of Education, Universidad Católica de Temuco, Temuco 4810296, Chile; 2International Observatory on School Climate and Violence Prevention (IOSCVP), 41004 Sevilla, Spain; 3Faculty of Social Sciences and Arts, Universidad Mayor, Temuco 4801043, Chile; 4Faculty of Psychology, Universidad Autónoma de Madrid, 28049 Madrid, Spain; 5Faculty of Philosophy and Humanities, Universidad Austral de Chile, Valdivia 5110566, Chile

**Keywords:** school climate, school coexistence, teacher well-being, victimization, cognitive processes, Chilean education

## Abstract

This study explores the perceptions of Chilean educational staff regarding school climate, classroom climate, and personal well-being, analyzing differences by gender and professional role. A non-experimental, cross-sectional, and quantitative design was used, with 8536 participants who completed perception scales on institutional support, classroom dynamics, and personal well-being. Results showed that women reported higher scores in Teacher–Student Relationship and Course Organization and Participation, while men reported higher levels of Institutional Support. Teachers reported more positive results than assistants in most dimensions, except for Institutional Support, where assistants scored higher. Although these effects were statistically significant, their magnitudes were consistently very small, underscoring the need for cautious interpretation. The findings highlight the importance of developing inclusive strategies that consider gender and role differences to foster positive and safe school environments. Limitations regarding the cross-sectional design, reliance on self-report measures, and the use of secondary data are acknowledged, and future research is suggested to explore cultural and structural factors that shape school coexistence.

## 1. Introduction

School violence has been understood as a complex phenomenon that transcends individual actions and reflects broader social dynamics ([18]; [64]). In the educational context, it takes physical, verbal, relational, and digital forms, affecting both the coexistence and well-being of students and teachers ([12]; [17]). This violence can become normalized in daily interactions, conditioning the perception of the school environment ([11]; [67]). From the Demands–Resources framework ([24]; [54]), school violence operates as a demand that deteriorates the school climate and classroom climate, while the latter function as resources that cushion its effects on teacher well-being.

Teachers’ perceptions of school violence are a critical aspect in understanding the dynamics of coexistence and the associated challenges in the educational environment. Teachers observe and face various forms of violence, including that between students, from students to teachers ([42]), which can affect both teaching–learning processes and teachers’ well-being. The development of specific instruments, such as the questionnaire developed by [42] ([42]), made it possible to capture these perceptions from a multidimensional perspective, assessing both the frequency and intensity of different types of violence in the Chilean context. This approach has revealed important correlations between the management of school coexistence and the reduction in violent acts, highlighting the role of clear institutional strategies and defined protocols to address conflicts ([46]).

Thus, the management of school coexistence focuses on conflict prevention and resolution, which implies the promotion of a positive school climate that favors learning and well-being. In this sense, interventions based on the collaboration of the entire educational community—teachers, students and families—have proven to be effective in mitigating the effects of violence ([2]). Furthermore, teachers’ perceptions of the effectiveness of these measures vary according to factors such as educational level, gender and type of institution, which highlights the need for differentiated and contextualized approaches ([63]).

Recent studies highlight that women teachers tend to report higher levels of violence among students but also evaluate more positively the management strategies implemented ([6]). These differences could be linked to cultural and gender factors that influence the interpretation and management of conflict situations. The study by [42] ([42]) showed that adequate management of coexistence has a significant effect. This finding reinforces the importance of integrating evaluation tools to monitor and continuously improve school practices. It also highlights the need to include training programs that strengthen the socioemotional competencies of teachers, facilitating the creation of safer and more equitable educational environments ([60]).

Teachers sometimes face direct aggression, which generates chronic stress and affects their capacity for emotional regulation ([17]). This phenomenon negatively impacts classroom management, increasing the perception of workload and reducing their job satisfaction. Similarly, the management of school coexistence becomes a critical factor in this context, since educational institutions that implement effective strategies to foster a positive climate manage to reduce the adverse effects of violence on teachers ([33]).

Notwithstanding the above, the lack of administrative support or the inexistence of clear policies aggravates emotional difficulties, especially in vulnerable environments ([12]). This emotionally demanding environment can contribute to the development of emotional dysregulation, manifesting itself in difficulties in controlling impulses, maintaining emotional clarity and adequately managing behavior in stressful situations ([20]). In addition, teachers’ previous experiences of adversity, such as a history of emotional abuse, may intensify their sensitivity to conflict situations, influencing their emotional responses in the classroom. This cycle of emotional stress compromises their well-being and also influences their ability to create a safe and positive learning environment for students ([19]). Therefore, it is essential to implement training programs in emotional regulation and self-care strategies that strengthen their resilience to the demands of the educational context.

Therefore, it is essential to implement training programs in emotional regulation and self-care strategies that strengthen their resilience to the demands of the educational context ([19]). In this sense, the school climate emerges as a fundamental factor that shapes these educational experiences ([34]). The school climate, a multidimensional construct that encompasses the quality and character of school life, plays a significant role in educational outcomes and student well-being ([9]; [45]). It includes factors such as safety, relationships, teaching practices, and the institutional environment that directly influence everyday school experiences ([59]). A positive school climate is associated with better academic performance, fewer school problems, and improved well-being among students and teachers ([9]; [59]). Among the elements that contribute to a positive climate are teacher–student relationships, peer interactions, and a sense of belonging. In particular, the classroom climate, which focuses on social interactions and expectations within each classroom, significantly influences the overall school climate ([5]).

Despite the importance of school climate, classroom climate, and personal well-being in educational settings, few studies examine how educators’ perceptions of these dimensions vary based on individual and contextual factors. While previous studies have explored these constructs independently, no research was found in the Chilean context that comprehensively analyzes how gender, professional role, and workplace location simultaneously influence the perceptions of school system employees. Understanding these differential perceptions is important for comprehending and explaining the specific needs of education staff, which will ultimately contribute to improving school coexistence and educational outcomes.

For all the reasons outlined above, this study analyzes the perceptions of Chilean educational staff regarding school climate, classroom climate, and personal well-being. The objective is to identify differences according to gender and professional role and to examine whether the combination of both variables modifies these perceptions.

## 2. Materials and Methods

The methodology used in this study follows a quantitative, non-experimental, cross-sectional design ([29]; [37]).

### 2.1. Participant

A total of 8536 people participated: 27.4% men and 72.6% women, 81.4% teachers and 18.6% education assistants. In the Chilean context, assistants are referred to as ‘asistentes de la educación’, a legally defined occupational category that includes professional, para-teaching, and support service staff. The sample included staff from schools across the national territory, within the Chilean school system. Years of experience ranged from 1 to 61 (M = 12.56; SD = 10.51). The surveys recorded participants’ ages in five-year intervals. This method, commonly used in educational and social research, serves to protect participants’ confidentiality, strengthen statistical analyses by consolidating sample sizes, and simplify the interpretation of results, thereby aiding in the effective identification of patterns and trends ([14]). Details on participants by age group, macro-zone, region, and municipality are provided in the Appendix A.

### 2.2. Instrument

The study used three separate instruments developed for the School Coexistence Monitoring Program (in Chile) by Junta Nacional de Auxilio Escolar y Becas ([27]). These instruments were developed as part of the FONDEF project IT14I10132 ‘Packaging and Transferring a School Coexistence Monitoring System to the JUNAEB Life Skills Public System Program’ within the broader Life Skills Program framework. No studies reporting the psychometric properties of these instruments were found in the literature; therefore, their reliability and validity are examined in the present study. Nonetheless, it should be noted that the classroom climate scale has previously been examined with Chilean students ([35]), showing adequate reliability and construct validity. These instruments were designed to explore the perceptions of educational personnel on three distinct dimensions: School Climate (SC), Classroom Climate (CC), and Personal Well-being PW:

School Climate (SC): This instrument evaluates teachers’ and education assistants’ perceptions of the institutional environment, including institutional support, participation opportunities, personal contribution, as well as experiences of victimization and perceptions of violence as a problem. It consists of 20 items.

Classroom Climate (CC): This instrument assesses aspects related to well-being in classroom dynamics, such as perceptions of the physical environment, teacher–student and student–student relationships, learning orientation, and classroom organization and participation. It contains 19 items.

Personal Well-being (PW): This unidimensional instrument reflects teachers’ and assistants’ perceptions of their motivation to teach and job satisfaction. It includes 10 items addressing factors related to professional performance and the work environment.

Each item is measured using a 4-point Likert-type scale, which makes it possible to identify levels of agreement and facilitate comparative analysis in different areas evaluated. In the Victimization and Fear subscale of the School Climate Scale, a higher score indicates greater exposure to threatening situations. In the other subscales, a higher score reflects a better perception of what is measured. Therefore, the Victimization and Fear subscale is expected to have negative correlations with the others.

### 2.3. Procedure

This study is based on a secondary analysis of data from the 2018 application, provided by the JUNAEB, the Chilean government agency responsible for administering the surveys of the Monitoring School Coexistence program. The researchers obtained the information through a formal request submitted in accordance with the Transparency Law for Public Function and Access to State Administration Information —Law No. 20285—([41]), as documented in exempt resolution DN-02620/2024. More recent applications of the survey were not provided to the researchers, as JUNAEB denied access on the grounds that the corresponding datasets had not yet been anonymized. This resolution explicitly granted access to the requested data, ensuring full compliance with Chilean legal requirements.

In exempt resolution DN-02620/2024, it is clarified that researchers did not have access to any sensitive or identifiable information, strictly complying with ethical standards and privacy guarantees. Furthermore, JUNAEB confirmed that participation in the survey was voluntary and that participants were informed of the study’s objectives and the confidentiality of the data as part of the informed consent process. This study shows how important it is to use publicly available data for research within a legal and transparent framework. In addition, using open government data in this research highlights its recognized role in advancing social and scientific progress ([48]).

For these reasons, the study complied with international ethical guidelines, including the Declaration of Helsinki and the Singapore Declaration, as well as Chilean legislation regarding the protection of personal data and the handling of sensitive information, such as Law No. 19628 on the Protection of Private Life and Law No. 20120 ([40]). These measures ensured the ethical use of data and reinforced the integrity and transparency of the research process. For transparency, the resolution authorizing access to the data (DN-02620/2024) is included in the Data Availability section of this manuscript.

### 2.4. Plan of Analysis

Cases with more than 20% missing data were eliminated in a highly stringent decision, even exceeding the minimum criteria proposed by [57] ([57]). This was done to reduce the proportion of imputed data and preserve as much directly observed information as possible. Missing data patterns were analyzed to determine the appropriate imputation method using Little’s MCAR Test ([31]) and tests for multivariate normality and homoscedasticity. Since the data did not meet assumptions of random distribution, normality, and equal variances, the missForest imputation method was used ([55]), resulting in the final database of 8536 cases reported in the participants section.

Given that there are no reports on the quality of the scales, the proposed structure was contrasted. To do so, and according to the criteria referred to by [25] ([25]) the normality of the items was evaluated by means of the Kolmogorov–Smirnov test, in order to determine the most appropriate methods for data analysis. Subsequently, a Confirmatory Factor Analysis (CFA) was carried out to estimate the degree of fit between the theoretical model and the data, using the polychoric correlation matrix and the unweighted least squares estimation method with adjusted mean and variance (ULSMV). Model fit was assessed using the Root Mean Square Error of Approximation (RMSEA), the Comparative Fit Index (CFI), and the Tucker–Lewis Index (TLI). Although chi-square statistics are reported, they are not used as primary indicators of fit due to their sensitivity to large sample sizes, and χ^2^/df ratios above 5 were likewise interpreted with caution, given their sensitivity to large sample sizes ([25]).

The reliability of the instrument was evaluated by means of composite reliability, through the McDonald omega coefficient ([36]), for which values greater than 0.65 are admissible, values greater than 0.7 are acceptable, between 0.8 and 0.9 are good and values equal to or greater than 0.9 are excellent ([1]). The convergent validity of each scale was evaluated considering that each subscale meets the following criteria: (1) the standardized loadings of the indicators must be greater than 0.5 and present a level of statistical significance with a *p*-value of less than 0.05; (2) the average variance extracted (AVE) values must exceed 0.5; and (3) the composite reliability must be greater than 0.7 ([23]). Discriminant validity was assessed by comparing the shared variance with the mean variance extracted. Evidence of discriminant validity is established when the square root of the AVE of a latent variable is greater than the correlations between that variable and the others ([16]).

In addition, for scales whose subscales do not reach discriminant validity, the analysis follows the approach of [30] ([30]), by checking the correlations between latent variables in a first-order confirmatory model. In this context, a correlation lower than 0.5 indicates discriminant validity, while values higher than 0.85 suggest convergent validity between factors, which may require the respecification of the structure, provided that the resulting model is significant. According to [62] ([62]), when high correlations are observed between first-order factors and the model is supported by specific theoretical concepts, it is possible to perform a second-order Confirmatory Factor Analysis (CFA). In this case, the first-order latent variables form one or more second-order factors, maintaining the assumptions of the first-level CFA measurement model.

Following methodological tradition ([7]; [38]; [61]), measurement invariance was tested through a hierarchical sequence of models: (1) the configural model, a multigroup CFA in which the same factor structure is specified across groups while allowing freely estimated loadings, intercepts, and error variances; (2) the metric model, which constrains factor loadings to be equal across groups; and (3) the scalar model, which additionally constrains intercepts (or thresholds in the case of ordinal items) to equality.

The evaluation of the different levels of measurement invariance was based on the comparative analysis of model fit indices across nested models. First, ΔCFI was considered, following the recommendation of [7] ([7]), who indicate that a deterioration of ≤0.01 suggests that the imposed restriction does not significantly worsen model fit. Complementarily, ΔRMSEA was examined, with values of deterioration up to 0.015 considered acceptable. It should be noted that these thresholds are applied only when model fit worsens after imposing restrictions; in cases where fit indices improve, this is interpreted as evidence consistent with the assumptions of invariance.

Having established this, group differences were examined using factorial ANOVA for each subscale, with gender and professional role as factors. Given the large sample size (*n* = 8536), this approach is robust to violations of normality assumptions. Differences were reviewed at a statistical significance level of 0.05 ([8]).

Finally, the cut-off points for the latent variables were determined by cluster analysis using the K-Means method on the average scores of the subscales. This approach allowed the identification of homogeneous groups in a simple and interpretable way, facilitating the practical application of the results for school readers. Although this method implies a simplification by using subscale averages, it was justified by the need to make the findings more accessible and applicable to educational contexts.

SPSS v.27 ([26]) was used to perform normality tests, hypothesis testing, descriptive statistics, and K-Means cluster analysis. Missing data analysis and imputation were performed using RStudio v. 2024.09.1+394 ([50]). CFA, measurement invariance analysis, and SEM analysis were performed with MPlus v.8.1 ([44]). For AVE, reliability, convergent and discriminant validity calculations, Excel v.16 ([39]) was used.

## 3. Results

Missing data analysis using Little’s MCAR Test indicated that data were not missing completely at random (*p* < 0.05). Tests for normality and homoscedasticity showed non-compliance with these assumptions. Consequently, the missForest imputation method was applied, retaining the 8536 cases reported in the Participant Section.

Both the School Climate and Classroom Climate scales showed high correlations among certain first-order latent variables; therefore, a second-order factor model was estimated for each instance (Table 1 and Table 2). In the case of the Personal Well-being scale, one indicator was eliminated because its content was not related to the other items, which strengthened the internal coherence of the construct (Table 3).

The goodness-of-fit indices show a good fit of the proposed model to the data in each of the three scales (Table 4).

The School Climate scale shows reliability, with McDonald’s omega coefficients reaching good and excellent levels. Likewise, convergent validity is observed, since all the standardized loadings of the indicators are greater than 0.5 and are statistically significant (*p* < 0.05), and the subscales present average variance extracted (AVE) values greater than 0.5. Finally, discriminant validity was not achieved for the Institutional Support and Participation subscales according to the Fornell–Larcker criterion, which further supported the decision to estimate a second-order model (Table 5).

The Classroom Climate scale shows high levels of reliability, with McDonald’s omega coefficients classified as good and excellent. In addition, it presents convergent validity, since all the standardized loadings of the indicators exceed 0.5, with statistical significance (*p* < 0.05), and the subscales reach mean variance extracted (AVE) values greater than 0.5. Finally, discriminant validity was not achieved for the Physical Environment, Student–Student Relationship, and Learning Organization subscales according to the Fornell–Larcker criterion, providing additional justification for estimating a second-order model (Table 6).

The unifactorial Personal Well-being scale shows high reliability, with McDonald’s omega coefficients at an excellent level. It also shows convergent validity, given that all the standardized loadings of its indicators exceed 0.5 and are statistically significant (*p* < 0.05). In addition, it presents an average variance extracted (AVE) value higher than 0.5, which supports the adequacy of the model (Table 7).

Measure invariance analyses confirmed that scalar invariance was achieved across all scales and categories (Table 8).

The means and standard deviations by gender and professional role are presented in Table 9, while the results of the factorial ANOVA are shown in Table 10.

The results of the factorial ANOVA revealed statistically significant effects on several dimensions of school climate. With regard to gender, statistically significant differences were found in Institutional Support (F = 7.856, *p* = 0.005, η^2^p = 0.001), where men reported a higher perception of support than women. Likewise, women scored significantly higher on Teacher–Student Relationship (F = 10.220, *p* = 0.001, η^2^p = 0.001) and Course Organization and Participation (F = 10.964, *p* < 0.001, η^2^p = 0.001). In contrast, women reported better Student–Student Relations (F = 18.228, *p* < 0.001, η^2^p = 0.002) and greater Personal Well-being (F = 9.063, *p* = 0.003, η^2^p = 0.001).

Teachers scored significantly higher than attendees in Satisfaction with School (F = 11.636, *p* < 0.001, η^2^p = 0.001), Teacher–Student Relationship (F = 104.107, *p* < 0.001, η^2^p = 0.012), Student–Student Relationship (F = 36.958, *p* < 0.001, η^2^p = 0.004), Course Organization and Participation (F = 31.772, *p* < 0.001, η^2^p = 0.004), and Personal Well-being (F = 27.326, *p* < 0.001, η^2^p = 0.003). Assistants reported significantly higher Institutional Support (F = 14.421, *p* < 0.001, η^2^p = 0.002). Although several of these effects reached statistical significance, the partial eta squared values were consistently below 0.01, indicating that the magnitude of the effects was very small.

A statistically significant interaction was identified in Course Organization and Participation (F = 4.725, *p* = 0.030, η^2^p = 0.001), indicating that the effect of gender on this dimension depends on professional role. The interaction can be seen in Figure 1.

The cut-off points for the latent variables, calculated by K-Means cluster analysis, are presented in Table 11, showing the homogeneous groups identified from the average scores of the subscales.

Based on the cut-off points, the percentage distribution of participants can be observed at different levels of perception for each scale (Table 12).

## 4. Discussions

The study analyzed how teachers’ and teaching assistants’ perceptions of school climate, classroom climate, and personal well-being in Chile vary according to gender and professional role. Understanding these different perceptions is important for designing strategies to improve school coexistence and the well-being of education staff, addressing inequalities and conflicts ([13]; [42]) and promoting safer and more equitable environments in educational communities.

In the Institutional Support dimension, statistically significant differences were found according to gender, with men reporting a higher perception of support than women. The relevance of this finding lies in the fact that institutional support is essential for the performance and well-being of educational staff. As [21] ([21]) has shown, effective institutional support strategies can reduce emotional exhaustion and improve professional performance, especially among teachers.

Gender differences in the perception of institutional support could be explained by various contextual and gender factors that influence professional interactions within the educational context. Previous research has identified that specific work dynamics can vary according to the gender of educational staff ([33]). Furthermore, during complex situations such as the COVID-19 pandemic, institutional support has been shown to have a positive impact on educators’ work–life balance, although this impact may be experienced differently ([28]).

In the European context, research in inclusive primary schools has highlighted the value of interprofessional collaboration, closely linked to institutional support for teamwork ([65]). Similarly, the differences in perceptions of institutional support according to gender found in this study may be related to variations in professional interactions and access to specific resources within Chilean educational institutions.

Women also obtained significantly higher scores in the dimensions of Teacher–Student Relationship and Course Organization and Participation, showing statistically significant differences in both areas. These findings are relevant considering that the quality of the teacher–student relationship is an important factor for students’ well-being, engagement, and academic performance, as previous studies have shown ([3]; [22]; [32]; [47]).

In terms of teacher–student relationships, women teachers report more positive perceptions of these interactions. Existing research reveals that positive relationships between teachers and students have a significant impact on well-being, engagement, and academic performance ([10]; [22]). This ability of teachers to build positive and meaningful relationships is reflected in the quality of educational interactions, as some studies have shown ([15]; [47]).

With regard to course organization and participation, gender differences in classroom dynamics and teacher perceptions have been explored recently, revealing varied findings depending on the educational level and context. For example, in physics classrooms, teachers report more gender differences than students themselves, especially in learning characteristics and teacher–student interactions ([43]). In primary education, specifically in science content, although teachers dominate in terms of speaking time, students participate with equal frequency, suggesting that teaching strategies to encourage social interaction are more relevant than gender stereotypes ([49]).

In contrast, women reported better student–student relationships and greater personal well-being, showing statistically significant differences in both dimensions. These findings reflect gender-differentiated patterns in the perception of interpersonal dynamics and well-being in the educational context, suggesting that the work experiences of educational staff are influenced by gender factors that require specific consideration in the design of institutional policies ([53]).

In terms of student–student relationships, women reported more positive perceptions of these interactions. These results are consistent with previous studies, which highlight the importance of interpersonal relationships in the educational environment. Research reveals that positive relationships between teachers and students, as well as among students themselves, have a significant impact on well-being, engagement, and academic performance ([10]; [22]). Women teachers, having greater opportunities to observe and evaluate interactions among students, may be better able to identify potential conflicts, foster collaboration, and promote a climate of mutual respect and support.

With regard to personal well-being, women reported significantly higher levels than men. In the area of personal well-being among teachers, research has shown that men teachers may report higher levels of well-being than their women counterparts in some contexts, although this may contradict findings of higher levels of stress among men ([52]; [66]). Work engagement has been identified as an essential mediator between the ratio of positive to negative emotions and overall well-being, highlighting the importance of fostering positive emotions in educational settings ([51]). Factors such as teaching experience and type of school also significantly influence well-being and perceived stress, with different patterns between private and public schools ([56]; [66]).

In the School Satisfaction dimension, teachers reported a more favorable score than education assistants. Teacher job satisfaction is linked to factors such as the work environment and interpersonal relationships, which can vary depending on the role and responsibilities within the school organization ([4]). Therefore, it is plausible that teachers, who have more central roles in the educational process, perceive a more favorable work environment, contributing to greater school satisfaction.

With regard to institutional support, assistants perceived significantly higher levels than teachers. Institutional support is essential for the performance and well-being of educational staff. Effective institutional support strategies can reduce emotional exhaustion and improve professional performance, especially among teachers ([21]). The difference in perceptions between teachers and assistants could be related to differences in professional interactions and access to specific resources depending on institutional role.

In the Teacher–Student Relationship and Student–Student Relationship dimensions, teachers reported more positive perceptions than attendees. Although considering the nature of these scales the result might seem obvious, this finding is relevant given that the quality of teacher–student relationships is an important factor for students’ well-being, engagement, and academic performance ([3]; [32]). Teachers, who play a central role in the educational process, have greater opportunities to establish meaningful relationships with students and observe group dynamics in a comprehensive manner.

In Course Organization and Participation, teachers also outperformed attendees. Research in primary science education suggests that teaching strategies to encourage social interaction are more relevant than other factors, which could explain why teachers, who have greater control over these strategies, perceive better organization and participation ([49]).

In Personal Well-being, teachers reported significantly higher levels than attendees. Research has identified that job commitment acts as a mediator between emotions and overall well-being, highlighting the importance of fostering positive emotions in educational settings ([51]). Differences in well-being between teachers and assistants could be associated with variations in perceived emotional demands or differential access to resources and support networks according to their specific roles.

A particularly relevant finding was the identification of a statistically significant interaction in the Course Organization and Participation dimension, indicating that the effect of gender in this dimension depends on the professional role. This result represents one of the most important findings of the study, as it suggests that gender differences do not operate uniformly across all educational roles but are modulated by the specific function that each person performs in the school context. The interaction observed reveals that patterns of perception of course organization and participation vary in complex ways depending on the combination of gender and professional role. This interaction shows that gender differences in the perception of course organization are not consistent between teachers and teaching assistants, suggesting that the context of the professional role modifies the way men and women experience and evaluate this dimension.

This finding is consistent with previous research that has explored gender differences in classroom dynamics and teacher perceptions. In preschool education settings, teachers’ gender perceptions are influenced by professional experience and participation in gender-oriented educational activities ([68]). Similarly, in the case of future secondary school teachers, gender also impacts perceptions of teaching, with women students retaining more negative memories of their previous teachers ([53]).

The presence of this interaction highlights the importance of adopting more nuanced approaches when analyzing gender differences in the educational context, recognizing that professional roles act as a significant moderator of these differences. This has important implications for the development of institutional strategies that consider gender differences and how they manifest differently according to specific roles within the educational system.

Given that JUNAEB does not provide specific interpretative frameworks for the results of its school coexistence monitoring instruments, cut-off points were established using K-Means cluster analysis for each dimension evaluated. These parameters are made available as normative references for educational contexts similar to Chile’s, allowing institutions to interpret their results more informedly in relation to educational staff perceptions of school climate, classroom climate, and personal well-being. It is important to note that these cut-off points should be considered as specific contextual references for the Chilean education system, and their applicability in other contexts should be evaluated considering the cultural and structural particularities of each system.

In addition to the cut-off points, the mean values on the 4-point Likert scales provide useful context. Personal Well-being scores were consistently above 3, reflecting positive perceptions, while Victimization and Fear averaged around 1.9, a relatively low score that nevertheless signals a concern. These values indicate that group differences emerged within a generally positive perception of climate and well-being, except for the area of victimization.

Practical interventions that strengthen teachers’ socioemotional competencies, promote collaborative work between teachers and assistants, and foster participatory classroom practices have been shown to improve school climate and staff well-being. Programs oriented toward social-emotional learning, peer collaboration, and institutional support mechanisms contribute to creating safer and more supportive environments for both students and education personnel (e.g., [58]).

## 5. Conclusions

This study reveals how Chilean education personnel’s perceptions of school climate, classroom climate, and personal well-being are influenced by gender and professional role, with differentiated and complex patterns in each dimension evaluated. The findings show that women report higher perceptions of teacher–student relationships and course organization, while men perceive greater institutional support. Teachers report more positive results than assistants in most dimensions, whereas assistants perceive higher levels of institutional support. Of particular note is the identification of a statistically significant interaction between gender and professional role in course organization and participation, showing that gender differences do not operate uniformly across all educational roles. It is also important to note that, although statistically significant, the effect sizes were consistently very small, which calls for cautious interpretation of the results. The findings highlight the importance of developing inclusive institutional strategies that consider gender and role differences not independently but together with complex interactions, in order to promote a positive school climate and reinforce the well-being of all educational staff as a priority in school settings.

This study has some limitations that should be considered. The cross-sectional design used does not allow causal relationships to be established between the variables evaluated, limiting inferences about the direction of the effects. Furthermore, the exclusive use of self-report measures may introduce bias in the responses due to the participants’ personal or social perceptions. Although the findings are relevant to the Chilean educational context, their generalization to other international educational systems may be limited due to cultural and structural differences.

At this point, it is worth noting that this study demonstrates the value of using public government data for scientific research, following open science principles that allow transparent access to information relevant to social and educational development. However, the use of open data also has limitations, particularly in terms of the control that researchers have over the study design, the quality of the data collected, and the variables included in the original databases. These limitations inherent in the use of secondary data should be considered when interpreting the results, although they do not diminish the value of leveraging existing public resources to generate scientific knowledge that contributes to the improvement of education systems. In addition, as the dataset is provided by an external government agency, it does not include school-level identifiers, which limits the possibility of conducting analyses at the institutional level.

The limitations identified in this study open up various opportunities for future research. First, it would be valuable to develop longitudinal studies to examine the temporal evolution of educators’ perceptions and establish causal relationships between school climate, classroom climate, and personal well-being variables. Such designs would provide a better understanding of the factors that influence changes in perception over time and identify complex moments in educators’ work experience.

Another limitation concerns the timing of the dataset, which corresponds to the 2018 application. Although it represents a large and nationally distributed sample, the use of older data may affect the immediate applicability of the findings to current school contexts. This limitation is compounded by the fact that JUNAEB did not provide access to subsequent applications of the survey, citing that the datasets had not yet been anonymized. Consequently, while the analyses remain valuable for understanding structural patterns in school climate and staff well-being, caution is warranted when extrapolating the results to more recent educational contexts.

The incorporation of mixed methodologies combining self-report measures with direct observations, in-depth interviews, and objective data from the school context could significantly enrich our understanding of these phenomena. It would also be relevant to explore the interactions between gender and professional role identified in this study through more detailed analyses that include additional moderating variables such as years of experience, students’ educational level, and socioeconomic characteristics of the school context.

Future research could also examine the replicability of these findings in other Latin American education systems, as well as develop international comparative studies to identify cultural and structural factors that influence the perceptions of education personnel, including variables that capture ethnic or cultural background, to explore potential differences in perceptions associated with these characteristics. Finally, it would be valuable to investigate the development and implementation of specific interventions based on the differentiated patterns identified, evaluating their effectiveness in improving the school climate and the well-being of education staff according to gender and professional role characteristics.

## Figures and Tables

**Figure 1 behavsci-15-01447-f001:**
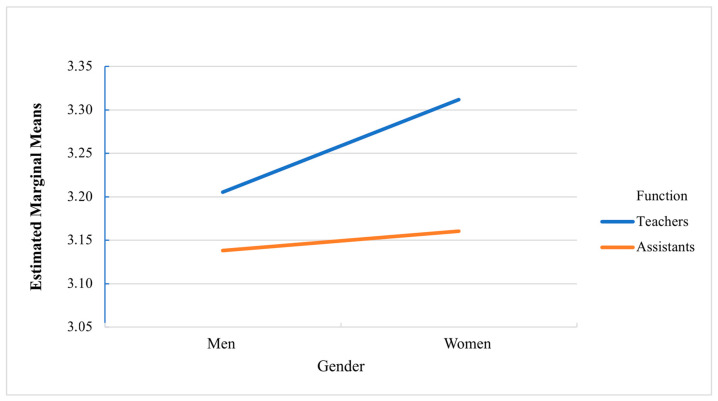
Interaction Course Organization and Participation. Source: Prepared by the authors.

**Table 1 behavsci-15-01447-t001:** Structure of the School Climate scale.

2nd-Order Factor	Factor	Included Items	Number of Items
School Climate (SC)	School Satisfaction (SS)	ce01–x06	6
	Victimization and Fear (VF)	ce07–x08	2
	Institutional Support (IS)	ce09–x14	6
	Participation (PA)	ce15–x18	4
	Personal Contribution (PC)	ce19–x20	2

Source: Prepared by the authors.

**Table 2 behavsci-15-01447-t002:** Structure of the Classroom Climate scale.

2nd-Order Factor	1st-Order Factor	Included Items	Number of Items
Classroom Climate (CC)	Perception of the Physical Environment (PPE)	ca01–ca04	4
	Teacher–Student Relationship (TSR)	ca05–ca06	2
	Student–Student Relationship (SSR)	ca07–ca12	6
	Learning Orientation (LO)	ca13–ca16	4
	Course Organization and Participation (COP)	ca17–ca19	3

Source: Prepared by the authors.

**Table 3 behavsci-15-01447-t003:** Structure of the Personal Well-being scale.

Factor	Included Items	Number of Items
Personal Well-Being (PW)	bp01–bp07; bp09–bp10	9

Source: Prepared by the authors.

**Table 4 behavsci-15-01447-t004:** Goodness-of-fit indices of the scales.

Scale	χ^2^	df	*p*-Value	RMSEA	CFI	TLI
SC	6177.634	165	<0.001	0.064	0.957	0.947
CC	9112.055	163	<0.001	0.080	0.935	0.934
PW	1246.365	26	<0.001	0.074	0.988	0.984

Note: RMSEA values <0.08 indicate good fit. CFI and TLI values >0.90 indicate good fit, >0.95 excellent fit. Source: Prepared by the authors.

**Table 5 behavsci-15-01447-t005:** Reliability indices for the five subscales of School Climate.

Factor	Loadings Min–Max	AVE	ω	SS	VF	IS	PA	PC
SS	0.678–0.860	0.656	0.919	0.810				
VF	0.885–0.890	0.788	0.881	–0.388	0.888			
IS	0.679–0.873	0.645	0.916	0.710	–0.421	0.803		
PA	0.781–0.829	0.643	0.878	0.707	–0.290	0.848	0.802	
PC	0.952–0.961	0.915	0.956	0.529	–0.211	0.469	0.567	0.957

Note: The diagonal contains the square root of the AVE. Below the diagonal are the correlations between factors. Source: Prepared by the authors.

**Table 6 behavsci-15-01447-t006:** Reliability indices for the five Classroom Climate subscales.

Factor	Loadings Min–Max	AVE	ω	PPE	TSR	SSR	LO	COP
PPE	0.681–0.884	0.626	0.869	0.791				
TSR	0.856–0.919	0.789	0.882	0.634	0.888			
SSR	0.776–0.873	0.673	0.925	0.708	0.870	0.820		
LO	0.708–0.859	0.598	0.855	0.635	0.763	0.888	0.773	
COP	0.820–0.859	0.705	0.877	0.593	0.705	0.806	0.808	0.839

Source: Prepared by the authors.

**Table 7 behavsci-15-01447-t007:** Reliability and validity indices—Personal Well-being.

Factor	Loadings Min–Max	AVE	ω
PW	0.531–0.925	0.716	0.957

Source: Prepared by the authors.

**Table 8 behavsci-15-01447-t008:** Tests of measurement invariance across gender, role, and work setting.

Scale	Category	Level	χ^2^	df	*p*	RMSEA	CFI	TLI	ΔCFI	ΔRMSEA
SC	Gender	Configural	4675.076	320	<0.001	0.056	0.951	0.942	–	–
SC	Gender	Metric	3202.917	320	<0.001	0.045	0.968	0.964	0.017	–0.011
SC	Gender	Scalar	2783.111	320	<0.001	0.039	0.973	0.972	0.005	–0.006
SC	Role	Configural	3419.059	320	<0.001	0.048	0.951	0.942	–	–
SC	Role	Metric	2450.094	320	<0.001	0.038	0.967	0.962	0.016	–0.010
SC	Role	Scalar	2667.576	320	<0.001	0.038	0.964	0.963	–0.003	0.000
CC	Gender	Configural	6189.846	284	<0.001	0.070	0.948	0.937	–	–
CC	Gender	Metric	2771.289	298	<0.001	0.044	0.978	0.975	0.030	–0.026
CC	Gender	Scalar	2158.885	331	<0.001	0.036	0.984	0.983	0.006	–0.008
CC	Role	Configural	5421.174	284	<0.001	0.065	0.940	0.927	–	–
CC	Role	Metric	2732.692	298	<0.001	0.044	0.971	0.967	0.031	–0.021
CC	Role	Scalar	2212.651	331	<0.001	0.036	0.978	0.977	0.007	–0.008
PW	Gender	Configural	1068.199	52	<0.001	0.068	0.987	0.982	–	–
PW	Gender	Metric	275.055	60	<0.001	0.029	0.997	0.997	0.010	–0.039
PW	Gender	Scalar	218.805	77	<0.001	0.021	0.998	0.998	0.001	–0.008
PW	Role	Configural	993.632	52	<0.001	0.065	0.986	0.980	–	–
PW	Role	Metric	278.292	60	<0.001	0.029	0.997	0.996	0.011	–0.036
PW	Role	Scalar	158.96	77	<0.001	0.016	0.999	0.999	0.002	0.002

Source: Prepared by the authors.

**Table 9 behavsci-15-01447-t009:** Descriptive Statistics by Gender and Professional Role.

Subscale (Subs)	Descriptives M (SD)			
	Teachers		Assistants	
	Men	Women	Men	Women
School Satisfaction (SS)	3.43 (0.55)	3.40 (0.56)	3.36 (0.61)	3.34 (0.57)
Victimization and Fear (VF)	1.86 (0.88)	1.95 (0.87)	1.90 (0.88)	1.89 (0.84)
Institutional Support (IS)	3.11 (0.62)	3.05 (0.64)	3.19 (0.63)	3.13 (0.61)
Participation (PA)	3.14 (0.61)	3.12 (0.60)	3.13 (0.64)	3.13 (0.60)
Personal Contribution (PC)	3.70 (0.51)	3.67 (0.53)	3.70 (0.56)	3.69 (0.54)
Perception of the Physical Environment (PPE)	3.12 (0.63)	3.10 (0.64)	3.12 (0.61)	3.09 (0.64)
Teacher–Student Relationship (TSR)	3.51 (0.50)	3.54 (0.51)	3.30 (0.58)	3.39 (0.56)
Student–Student Relationship (SSR)	3.34 (0.48)	3.41 (0.49)	3.23 (0.56)	3.31 (0.52)
Learning Orientation (LO)	3.25 (0.54)	3.30 (0.53)	3.29 (0.56)	3.30 (0.52)
Course Organization and Participation (COP)	3.21 (0.57)	3.31 (0.56)	3.14 (0.64)	3.16 (0.58)
Personal Well-being (PW)	3.52 (0.45)	3.56 (0.43)	3.43 (0.52)	3.49 (0.45)

Source: Prepared by the authors.

**Table 10 behavsci-15-01447-t010:** Factorial ANOVA Results by Gender and Professional Role.

Subs	Gender				Function				Interaction			
	F	*p*	η^2^p	1−β	F	*p*	η^2^p	1−β	F	*p*	η^2^p	1−β
SS	1.607	0.205	<0.001	0.245	11.636	<0.001	0.001	0.927	0.001	0.971	<0.001	0.050
VF	1.427	0.232	<0.001	0.223	0.103	0.749	<0.001	0.062	2.520	0.112	<0.001	0.355
IS	7.856	0.005	0.001	0.800	14.421	<0.001	0.002	0.967	<0.001	0.999	<0.001	0.050
PA	0.147	0.702	<0.001	0.067	0.001	0.982	<0.001	0.050	0.230	0.631	<0.001	0.077
PC	0.663	0.415	<0.001	0.129	0.504	0.478	<0.001	0.109	0.347	0.556	<0.001	0.091
PPE	1.546	0.214	<0.001	0.237	0.087	0.768	<0.001	0.060	0.003	0.960	<0.001	0.050
TSR	10.220	0.001	0.001	0.892	104.110	<0.001	0.012	1	3.030	0.082	<0.001	0.413
SSR	18.228	<0.001	0.002	0.990	36.958	<0.001	0.004	1	0.019	0.892	<0.001	0.052
LO	2.797	0.094	<0.001	0.387	1.103	0.294	<0.001	0.183	1.140	0.286	<0.001	0.187
COP	10.964	<0.001	0.001	0.912	31.772	<0.001	0.004	1	4.725	0.030	0.001	0.585
PW	9.063	0.003	0.001	0.853	27.326	<0.001	0.003	0.999	0.336	0.562	<0.001	0.089

Source: Prepared by the authors.

**Table 11 behavsci-15-01447-t011:** Cut-off points by K-Means analysis.

Scale	Subscale	Low	Medium	High
	School Satisfaction	1.73	2.91	3.73
	Victimization and Fear	1.13	2.35	3.79
SC	Institutional Support	2.03	2.90	3.71
	Participation	1.96	2.85	3.62
	Personal Contribution	1.03	2.95	3.95
	Perception of the Physical Environment	1.98	2.90	3.81
	Teacher–Student Relationship	1.63	2.97	3.85
CC	Student–Student Relationship	2.24	3.05	3.81
	Learning Orientation	2.01	2.96	3.79
	Course Organization and Participation	1.76	2.87	3.74
PW	Personal Well-being	1.48	3.13	3.86

Source: Prepared by the authors.

**Table 12 behavsci-15-01447-t012:** Distribution of participants at each level of perception.

Scale	Subscale	Low	Medium	High
	School Satisfaction	3.9	31.5	64.6
	Victimization and Fear	45.3	46.4	8.3
SC	Institutional Support	14.9	46.2	38.9
	Participation	10.7	41.0	48.3
	Personal Contribution	1.1	23.6	75.3
	Perception of the Physical Environment	12.2	53.1	34.6
	Teacher–Student Relationship	1.5	36.6	62.5
CC	Student–Student Relationship	5.1	47.5	47.4
	Learning Orientation	4.0	51.5	44.4
	Course Organization and Participation	3.5	47.6	48.9
PW	Personal Well-being	0.7	42.5	56.7

Source: Prepared by the authors.

## Data Availability

The dataset used in this research was obtained from the Junta Nacional de Auxilio Escolar y Becas (JUNAEB), Chile, under Resolución Exenta No. DN-02620/2024. The information was provided in an entirely anonymized format. Data access for other researchers can be formally requested from JUNAEB in accordance with the Chilean Transparency Law (Law No. 20285). Moreover, a secondary dataset derived from the imputation of missing values and used for the present analyses is publicly available at https://doi.org/10.7910/DVN/KZ8B45 (accessed on 1 July 2025).

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
