# Peer review of "Gender and Professional Role Differences in Chilean Educational Personnel’s Perceptions of School Climate and Well-Being"

_behavsci, 2025, doi:10.3390/bs15111447_

Round 1
Reviewer 1 Report
Comments and Suggestions for Authors
Gender and Professional Role Differences in Chilean Educational Personnel's Perceptions of School Climate and Well-Being
This study explores the perceptions of Chilean educational personnel on school climate, classroom climate, and personal well-being, analyzing differences by gender, role, and workplace.
Regarding the theoretical framework:
This section should be reformulated and completed. Its writing should be conceived from an integrated ecological perspective, where the construct of school violence is developed, as the text begins with a brief definition: "School violence, understood as the intentional use of force to cause physical or psychological harm (WHO, 2014)." This definition is correct but very general and not specific to the school environment or its typologies (with typologies and levels: physical/psychological/relational, in-person/digital; classroom–school–community–policy). The above serves to explicitly articulate with the dimensions of School Climate (SC) —institutional support, participation, contribution, safety—, Classroom Climate (CC) —relationships, organization, learning orientation, physical environment— and Personal Well-being (PH) —motivation and job satisfaction—. In this sense, theoretically and in the analysis, the theoretical framework should make explicit that school violence operates as a demand on the educational system that deteriorates School Climate (SC) and Classroom Climate (CC): and that both act as resources that mediate or buffer its effect on Personal Well-being (PW): (Demands-Resources framework), cautiously incorporating differences by role and gender (more observations on this point in the methods section) and only if there is measurement equivalence. This reformulation requires aligning objectives, variables and inferences, linking each subscale with its function in the model (what it captures and what it does not), and reporting results with an emphasis on magnitude and precision (effect sizes and confidence intervals), so that theory, measurement and empirical evidence are coherently articulated. Likewise, the theoretical framework should develop the idea of coexistence or coexistence management in order to articulate all the scales' input regarding teaching staff perceptions. Without a robust conceptual framework that connects all the constructs, raises hypotheses, and coherently connects all sections of the material, this manuscript cannot be considered for publication.
Regarding materials and methods:
The authors note: "It was decided not to perform a measurement invariance analysis in this study (p. 225)" and base their rationale on Robitzsch & Lüdtke (2023). First, Robitzsch & Lüdtke (2023) is a recent article that points out that if invariance is not found, other alternatives are available, but they do not rule them out. The “classic” literature and current guidelines require invariance to compare means (Meredith, 1993; Vandenberg & Lance, 2000; Putnick & Bornstein, 2016) and explain that scalar invariance is what enables meaningful latent mean comparisons. The “it’s not essential” argument is irrelevant and does not authorize ignoring the problem. In this vein, Robitzsch & Lüdtke (2023) discuss scenarios where not requiring strict invariance may be reasonable, but they do not say “ignore it and do ANOVA”; they propose alternative approaches and warn of conditions. Otherwise, the results and conclusions may change after correcting for non-invariance. And this can happen in this scenario: comparing men vs. women and teachers vs. teaching assistants without testing measurement invariance (or without using an alternative method that models it) can confuse differences in the instrument with real differences in the construct. Therefore, I believe it is necessary for the editor to refer this article to a statistician.
Regarding results, discussion, and conclusions:
It is not possible to make suggestions in these sections, as the results are unreliable.
It should be noted that the authors are committed to exploring Chilean educational staff's perceptions of school climate, classroom climate, and personal well-being, analyzing differences by gender, role, and workplace. The latter, workplace, is not disputed. It should also be noted that the emphatic language used in the discussion should be moderated in light of the results, especially when the effects are minimal and invariance analysis is lacking.
Regarding English:
The English requires revision. The text is understandable, but there are issues with register ("Moreover/Additionally" instead of "Plus"), terminology (education/teaching assistants), table labels (Gender, Professional role, Subscale; p < .001), and punctuation/scientific style, among others.
Regarding English:
The English requires revision. The text is understandable, but there are issues with register ("Moreover/Additionally" instead of "Plus"), terminology (education/teaching assistants), table labels (Gender, Professional role, Subscale; p < .001), and punctuation/scientific style, among others.
Author Response
Response to Reviewer 1
We sincerely appreciate your comments, which allowed us to strengthen the manuscript both conceptually and methodologically. In relation to your observation on the need to broaden the conceptual framework and to consider school violence as a relevant phenomenon for understanding school climate and coexistence, we have added a new paragraph in the introduction. This section situates school violence as a factor that affects the educational climate and emphasizes the importance of analyzing both experiences of victimization and the positive dimensions that promote coexistence and well-being.
Some of the modifications were limited or combined with other suggestions, as in certain cases the reviewers expressed opposing views. We therefore hope it is clear that we have both added and removed content where necessary, seeking to balance the different comments. In addition, we corrected errors identified by the reviewers as well as others detected by the authors during this revision.
With regard to the measurement invariance analysis, in the initial version we followed the recommendation of a senior and widely recognized methodologist, who advised not to perform this procedure. However, as the lead author, I acknowledge that I did not fully understand how to justify that decision. Considering your remarks, it seemed more appropriate to conduct the full invariance analysis, which we have now incorporated following the standard hierarchical sequence (configural, metric, and scalar), in line with the specialized literature (Meredith, 1993; Vandenberg & Lance, 2000; Cheung & Rensvold, 2002). We are grateful for this observation, which has substantially strengthened the methodological rigor of the study.
Regarding discriminant validity, we carefully reviewed the scales using the Fornell–Larcker criterion. As you noted, in Table 6 it was necessary to present the correlation matrix, which confirmed that certain subscales did not meet discriminant validity. This became an additional element supporting the decision to estimate a second-order model, given the high correlations observed among some latent variables and the theoretical coherence of such a specification. During this review, we also detected a transfer error in the AVE column of Table 5, which affected the calculation of the diagonal in the Fornell–Larcker criterion. This error has been corrected in the revised version. The correction confirmed that discriminant validity was likewise not achieved for that scale, which further supported the decision to re-specify the model with a second-order factor.
We hope that these revisions have addressed your concerns, and we would be glad to respond to any further questions or clarifications you might consider necessary.
Reviewer 2 Report
Comments and Suggestions for Authors
Brief summary: The article presents a quantitative study that examines the individual and interactional effects of gender and professional role on well-being, drawing on a comprehensive analysis of school and classroom climates. The article is primarily relevant to consider in the Chilean educational context.
General concept comments:
The article is well-structured and asserts a relevant research topic in educational well-being from the perspective of professionals.
The abstract is clear and concise, considering the study's purpose, methodological approach, main results, and conclusions. The relationship between the object of study and the theory is unclear, and the research gap is not addressed – if considered, this could potentially amplify interest in readers.
The introduction highlights the dimension of school violence. Given the abstract and the methodology, even admitting the relation of school violence with school climate and overall well-being, school violence is far from the essence of the research: it is referred to as an element of the variable “Victimization and fear”, but nothing else is said on this topic in the results, discussion, and conclusion. There is a disengagement between theory and the rest of the article. Define the research questions/ hypotheses of your research, which are open-ended, as they can provide a better focus for the article. The gap in research is well-explored in the introduction.
The methodological design is rigorous and adequate, with options well supported in the literature. The sample size is very robust. Clarify information regarding the participants' origin and sampling, such as public/private schools, the entire country of Chile, or specific areas of provenance…; this is important to perceive the generalizability of the findings. Explain what is meant by assistants – other teachers, psychologists, other educational technicians, support staff…
The results are conveyed straightforwardly and clearly and are presented concisely. Regarding the model fit (Table 4), provide explanations for the X2/df, which are quite above the threshold of 5 (page 6, lines 265 and 266). Tables 5 and 6 regarding the scales in the analysis appear to be inconsistent – one includes the correlations between the subscales, while the other does not. Explain why not, or provide the correlations.
Regarding the results (Table 9), present an explanation regarding the values of partial eta squared because it goes beyond p-values, and tell the readers about the magnitude of the effect measured. For instance, for COP, even though (according to ANOVA) the differences between genders are significantly reliable, the magnitude is small (≤0.4% of variance explained) – so it is a significant but tiny effect (the effect is trivial). Similar reasoning should be considered for other variables. Review the discussion and conclusions in light of this approach.
The results analysis on page 8 presents an error – TSR and COP regarding gender (women score higher than men) and IS regarding professional role (assistants score higher) – it is said the opposite. Correct discussion and conclusions regarding these subjects.
Discussion should also consider the constructs’ mean values, taking into account the 4-point Likert scale used. The scores suggest no neutral overall well-being in most variables (mean values are above 3, and some above 3.5 - so positive). “Victimization and fear” is around 1.9, which is low but still concerning (it is something that warrants further analysis). This approach is necessary to characterize the perceptions of Chilean educational personnel.
Specific comments:
On page 1, line 14, specify professional role.
On page 1, line 19, the idea “teachers outside the classroom” is not aligned and comprehensive regarding the overall manuscript. Are these teachers the assistants?
On page 8, correct the typos in the table 9 heading.
Author Response
Response to Reviewer 2
We are grateful for your detailed and constructive observations, which guided us to improve the clarity, precision, and consistency of the manuscript. In some instances, your comments were not fully aligned with those of the other reviewers. For this reason, we carefully considered all perspectives, and when faced with opposing suggestions we sought a balanced solution, which can be seen in the tracked changes to the manuscript.
Regarding the analysis of results, we corrected the interpretation errors you pointed out in the Teacher–Student Relationship, Course Organization and Participation, and Institutional Support dimensions. The text now accurately reflects the descriptive statistics, with women scoring higher in the Teacher–Student Relationship and Course Organization and Participation dimensions, and assistants reporting higher Institutional Support compared to teachers. We sincerely thank you for noting these inaccuracies, which have now been amended.
In line with your request, we also emphasized the interpretation of partial eta squared values, making clear that although the effects were statistically significant, their magnitudes were consistently very small and therefore must be interpreted with caution. Similarly, we integrated a brief discussion of the mean values on the 4-point Likert scales, highlighting that Personal Well-being values were consistently above 3 (positive perception), while Victimization and Fear averaged around 1.9 (a low score but still relevant). These additions provide important descriptive context to complement the statistical findings.
You also suggested clarifying the term assistants. We addressed this by explicitly noting in the Participants section that in the Chilean context, assistants refers to ‘asistentes de la educación’, a legal category that includes professional, para-teaching, and support service staff. At the same time, we removed the previous distinction of “inside/outside the classroom,” as this variable was not analyzed and could generate unnecessary confusion. We believe this improves both the clarity and accuracy of the manuscript.
In addition, you raised a valid point regarding the origin of the participants. We clarified in the Participants section that the sample included educational staff from schools across the national territory, covering public, subsidized, and private institutions. It is important to note that the database provided by JUNAEB is anonymized and does not include direct identifiers of schools, which prevents us from reporting details at the institutional level. The only available geographical information corresponds to communes, from which broader regions and macro-zones can be inferred. However, this dataset is not of a censal type and does not achieve regional representativeness, so it cannot be used to project results at that level. For this reason, we chose to interpret the findings at the national school system level, while avoiding disaggregated information that could give a misleading impression of representativeness. We also added a note in the Limitations section clarifying that, as the dataset is provided by an external government agency, it does not include school-level identifiers, which restricts the type of analyses that can be conducted.
Nevertheless, we have presented all this information, which we believe would unnecessarily lengthen the manuscript, in a series of tables as supplementary material. This will allow you to judge whether it is sufficient to keep it in this section or whether it should be incorporated into the manuscript itself.
Finally, we standardized the tables in line with your observations. Specifically, we corrected the header from “Genre” to “Gender” and harmonized the presentation of thresholds in the measurement invariance tables (now Tables 8–10). In addition, following your indication, we clarified the evaluation of model fit, explicitly noting that chi-square and χ²/df ratios were considered inflated by the large sample size, while RMSEA, CFI, and TLI were prioritized as recommended in the literature.
We hope these revisions address your concerns and strengthen the overall contribution of the study. Please know that we would be glad to respond to any further questions or clarifications you may consider necessary.
Reviewer 3 Report
Comments and Suggestions for Authors
The study contributes to the literature by simultaneously analyzing gender and the professional role in the perception of school climate.
First, in the description of the instruments, I suggest including some examples of items for each scale.
Line 242 mentions testing hypotheses, yet the paper lacks a clear formulation of these.
The approach to the educational policies at the end is interesting, but it's worth providing a concrete example of an effective intervention.
Although the statistical significances are robust, the magnitude of the effect size is very small. An analysis in this regard can be briefly introduced.
Line 270. The School Climate....A word is missing (high). Reliability is not labeled.
Minor aspects:
Table 9 and Figure 1: The term "Genre" is very likely an editing error
In tables 8, 9, and 10, threshold values are rendered in different styles. There should be uniformity in the style of presentation.
The term "outside the classroom" is rarely used in articles to refer to teachers; instead, the phrase "out-of-classroom teachers" is more common.
Author Response
Response to Reviewer 3
We are grateful for your thoughtful and constructive review, which helped us improve both the clarity and the practical contribution of the manuscript. As with the other reviewers, some of your observations differed from theirs, and in those situations we carefully sought balanced solutions. We hope you will be able to see this process reflected in both the additions and the removals made in the revised manuscript.
In line with your observation regarding the lack of explicitly stated hypotheses, we revised the introduction to present the research objectives clearly and directly, without formulating formal hypotheses. We considered this approach more coherent with the exploratory design of the study and more consistent with the expectations of the other reviewers, while still addressing your concern about clarity in the research aims.
We also responded to your valuable point about effect sizes. Following your suggestion, we emphasized in both the Results and the Discussion that although several effects reached statistical significance, the partial eta squared values indicated that their magnitudes were consistently very small. This addition tempers the interpretation of the findings and aligns the conclusions more closely with the strength of the evidence.
You also recommended including practical examples of effective educational interventions. To address this, we added a short paragraph in the Discussion highlighting interventions that strengthen socioemotional competencies, promote collaborative work, and encourage participatory practices. These examples provide a concrete link between our findings and their possible implications for practice.
Finally, we standardized the tables and terminology across the manuscript, corrected errors noted by the reviewers, and identified and corrected some additional inconsistencies ourselves (for instance, in the reporting of discriminant validity and in table labeling). We also introduced the measurement invariance analysis and refined the evaluation of measurement models, which strengthens the methodological rigor of the study and responds to the concerns raised collectively by the reviewers.
We hope these revisions have adequately addressed your concerns, and we would be glad to provide any further clarifications if needed.
Round 2
Reviewer 2 Report
Comments and Suggestions for Authors
The suggested amendments were developed exhaustively by the authors, improving the clarity and rigour of the study documented.
The quantitative study used inferential analysis, which includes hypothesis testing. However, the manuscript does not clearly define the hypotheses being tested — only that group differences were examined (page 6, line 249). In other words, the hypothesis is implicit rather than explicit, as recommended by best practices.